# Highly efficient all-inorganic perovskite solar cells with suppressed non-radiative recombination by a Lewis base

Jing Wang[1,6], Jie Zhang[1,6], Yingzhi Zhou[2], Hongbin Liu [3], Qifan Xue[2], Xiaosong Li[3], Chu-Chen Chueh [4], Hin-Lap Yip [2]*, Zonglong Zhu[1]* & Alex K.Y. Jen[1,3,5]*

All-inorganic perovskite solar cells (PVSCs) have drawn increasing attention because of their outstanding thermal stability. However, their performance is still inferior than the typical organic-inorganic counterparts, especially for the devices with p-i-n configuration. Herein, we successfully employ a Lewis base small molecule to passivate the inorganic perovskite film, and its derived PVSCs achieved a champion efficiency of 16.1% and a certificated efficiency of 15.6% with improved photostability, representing the most efficient inverted all-inorganic PVSCs to date. Our studies reveal that the nitrile (C-N) groups on the small molecule effectively reduce the trap density of the perovskite film and thus significantly suppresses the non-radiative recombination in the derived PVSC by passivating the Pb-exposed surface, resulting in an improved open-circuit voltage from 1.10 V to 1.16 V after passivation. This work provides an insight in the design of functional interlayers for improving efficiencies and stability of all-inorganic PVSCs.

[1] Department of Chemistry, City University of Hong Kong, Tat Chee Avenue, Kowloon, Hong Kong. [2] Institute of Optoelectronic Materials and Devices, State Key Laboratory of Luminescent Materials and Devices, South China University of Technology, Guangzhou, China. [3] Department of Chemistry, University of Washington, Seattle, WA 98195, USA. [4] Department of Chemical Engineering, National Taiwan University, Taipei 10617, Taiwan. [5] Department of Materials Science and Engineering, City University of Hong Kong, Tat Chee Avenue, Kowloon, Hong Kong. [6]These authors contributed equally: Jing Wang, Jie Zhang. *email: msangusyip@scut.edu.cn; zonglzhu@cityu.edu.hk; alexjen@cityu.edu.hk

Currently, the certified power conversion efficiency (PCE) of organic–inorganic hybrid perovskite solar cells (PVSCs) has exceeded 25%, revealing very promising potential for commercial applications[1]. However, to realize commercialization, the issues of instability under thermal and light illumination stresses still need to be addressed, which mainly results from the high volatility of hydrophilic organic cations (e.g., $CH_3NH_3^+$) in the hybrid perovskite framework under such external stimuli[2,3]. To overcome this deficiency, partial or complete replacement of the volatile organic cations has been proven as an effective approach to improve device stability[4,5]. Very recently, all-inorganic perovskites with $Cs^+$ as the A-site cation ($CsPbI_xBr_{3-x}$, $x$ is in the range from 0 to 3) have drawn significant attention due to their outstanding thermal stability[6,7].

As a matter of fact, numerous efforts have already been dedicated to improve the performance of inorganic $CsPbI_xBr_{3-x}$ PVSCs, such as the utilization of quantum dots[8,9], compositional optimization[10,11], dimensional control[12,13], and interfacial engineering[14,15]. For example, a solvent-controlled growth method was recently developed to produce high-quality black-phase $CsPbI_3$ perovskite film and its derived PVSC can deliver a certified PCE of 14.67%[16]. Soon after, a $CsPbI_3$ PVSC with an impressive PCE of 17.06% was reported through the surface termination of the perovskite film by phenyltrimethylammonium bromide[17]. More recently, the PCE of $CsPbI_3$ PVSC has been improved to 18.4% by an integrated optimization of interface engineering and perovskite passivation[6]. Although the PCEs of inorganic PVSCs still fall behind those of the hybrid counterparts, these important works validate the promise of $CsPbI_xBr_{3-x}$ PVSCs.

It is noteworthy that most of the high-performance inorganic PVSCs reported so far are based on using the conventional n-i-p architecture, where a doped hole-transporting layer (HTL), like Spiro-OMeTAD, is employed. However, severe degradation processes have been reported for devices using such doped HTLs due to the instability of dopants (e.g., 4-tert-butyl pyridine and lithium salts)[18–21]. In this regard, the development of inverted p-i-n inorganic PVSCs is very attractive, since it not only can exempt the use of these unstable doped HTLs, but also be more compatible with the fabrication of tandem solar cells because most of perovskite–perovskite tandem solar cells are fabricated with the p-i-n architecture to date[22,23]. Therefore, it is highly desirable to realize high-efficiency inverted PVSCs. On one hand, it can enable the derived device to possess better operational stability; on the other hand, it can facilitate the development of perovskite–perovskite tandem solar cells to further enhance PCE.

Currently, there has been some studies reporting inverted all-inorganic PVSCs. For example, Han et al. reported a method of stabilizing cubic phase $CsPbI_3$ by systematically controlling the distortion of $[PbI_6]^{4-}$ in the lattice through using a series of organic terminal groups to achieve a stable PCE of 13.2% for the derived inverted PVSC[24]. Meanwhile, Mai et al. have employed a ZnO/C60 bilayer electron-transporting layer (ETL) to fabricate an inverted $CsPbI_2Br$ PVSC delivering a decent PCE of 13.3%[25]. These case manifested that such bilayer ETL can enhance carrier extraction and suppress leakage loss in the device. Later, the PCE of such inverted $CsPbI_2Br$ PVSC was further enhanced to 13.74% by introducing $InCl_3$ into the perovskite, which was the highest PCE reported for the inverted inorganic PVSCs until now[26]. However, the performance of all these inverted inorganic PVSCs developed are still significantly lagging behind those of their conventional counterparts, which is mainly restrained by the severe open-circuit voltages ($V_{OC}$) loss.

It has been reported that the positively/negatively charged defects on perovskite surface and grain boundaries can induce large $V_{OC}$ losses in PVSCs[27,28]. To address this issue, numerous works have been performed to passivate the perovskite surface/ grain boundaries, such as using Lewis base cations[29,30], polymer[14,31], and quaternary ammonium cations[32].

Inspired by these works, we present a simple molecular passivation strategy to reduce $V_{OC}$ loss by reducing the density of surface defects in $CsPbI_xBr_{3-x}$ film with $\pi$-conjugated 6TIC-4F with strong electron-donating core terthieno[3,2-b]thiophene (6 T) and two electron-withdrawing units 2-(5,6-difluoro-3-oxo-2,3-dihydro-1 H-inden-1-ylidene)malononitrile (IC-2 F)[33], which can be dissolved in antisolvent (such as chlorobenzene, CB) to passivate uncoordinated defects on surface/grain boundaries and facilitate charge extraction, further increase the photovoltaic performance. Theoretical calculations and experimental characterizations revealed that the numerous nitrogen (N) atoms possessing lone pair electrons on 6TIC-4F could passivate the surface defects of $CsPbI_xBr_{3-x}$ film via direct coordination with the lead ion ($Pb^{2+}$) through the formation of Lewis adducts, thereby suppressing the non-radiative recombination in the derived PVSC. Meanwhile, the employed 6TIC-4F tends to trigger the nucleation of perovskite precursor, leading to the formation of larger grain size and denser film. Furthermore, 6TIC-4F possesses a lowest unoccupied molecular orbital (LUMO) level of about −4.14 eV, which sits between the conduction band minimum (CBM) of perovskite (−3.49 eV) and ZnO (−4.25 eV)[25,34]. It enables a better energy alignment across the perovskite/ETL interface to provide improved electron extraction efficiency. Consequently, the as-optimized device with the structure of ITO/ $NiO_x$/$CsPbI_xBr_{3-x}$/ZnO/C60/Ag can deliver a remarkable PCE of 16.1% and a certified value of 15.6%, representing the best inverted all-inorganic PVSCs reported thus far.

## Results

**Perovskite structural and optical properties.** 6TIC-4F (Supplementary Fig. 1) is a low-bandgap acceptor reported by our group in previous work[33]. As illustrated in Fig. 1a, in this study, the $CsPbI_xBr_{3-x}$ films were prepared by one-step spin-coating method by adding dropwise 130 μL solution of 6TIC-4F in CB (3 mg mL$^{-1}$) onto the precursor film during the last 15 s of the spin-coating process. Afterward, the films were first annealed at 55 °C for 50 s, followed by annealing at 255 °C for another 50 s to get the black-phase perovskite. It needs to be pointed out that the thermal gravimetric analysis results in Supplementary Fig. 11 showed that neither 6TIC-4F nor 6TIC-4F/$CsPbI_xBr_{3-x}$ decompose in the range between 200 to 300 °C, indicating that the annealing condition in this work (255 °C) did not damage 6TIC-4F. Moreover, it has been reported that the perovskite—fullerene graded heterojunction could be formed within the perovskite film when the fullerene containing antisolvent was used during the film preparation[35–37]. We speculate that a perovskite—6TIC-4F graded heterojunction should also be formed within the $CsPbI_xBr_{3-x}$ film similar to those of the fullerene cases. This implies that there is a gradient distribution of 6TIC-4F within the heterojunction and the highest concentration should be on top of the film surface. In addition, due to the large size of the 6TIC-4F molecules, they should be located on the surface and grain boundaries of $CsPbI_xBr_{3-x}$ film, because they cannot be incorporated into the perovskite crystalline frame.

The X-ray diffraction method was conducted to study the perovskite crystal structure of the prepared films without and with 6TIC-4F treatment. As shown in Fig. 1b, both films showed apparent diffraction pattern peaks corresponding to the (100) and (200) lattice planes. Compared to the control film, the film with 6TIC-4F passivation had higher peak intensity, indicating 6TIC-4F could improve the film crystallinity without inducing structural changes. Grazing incidence wide-angle X-ray scattering

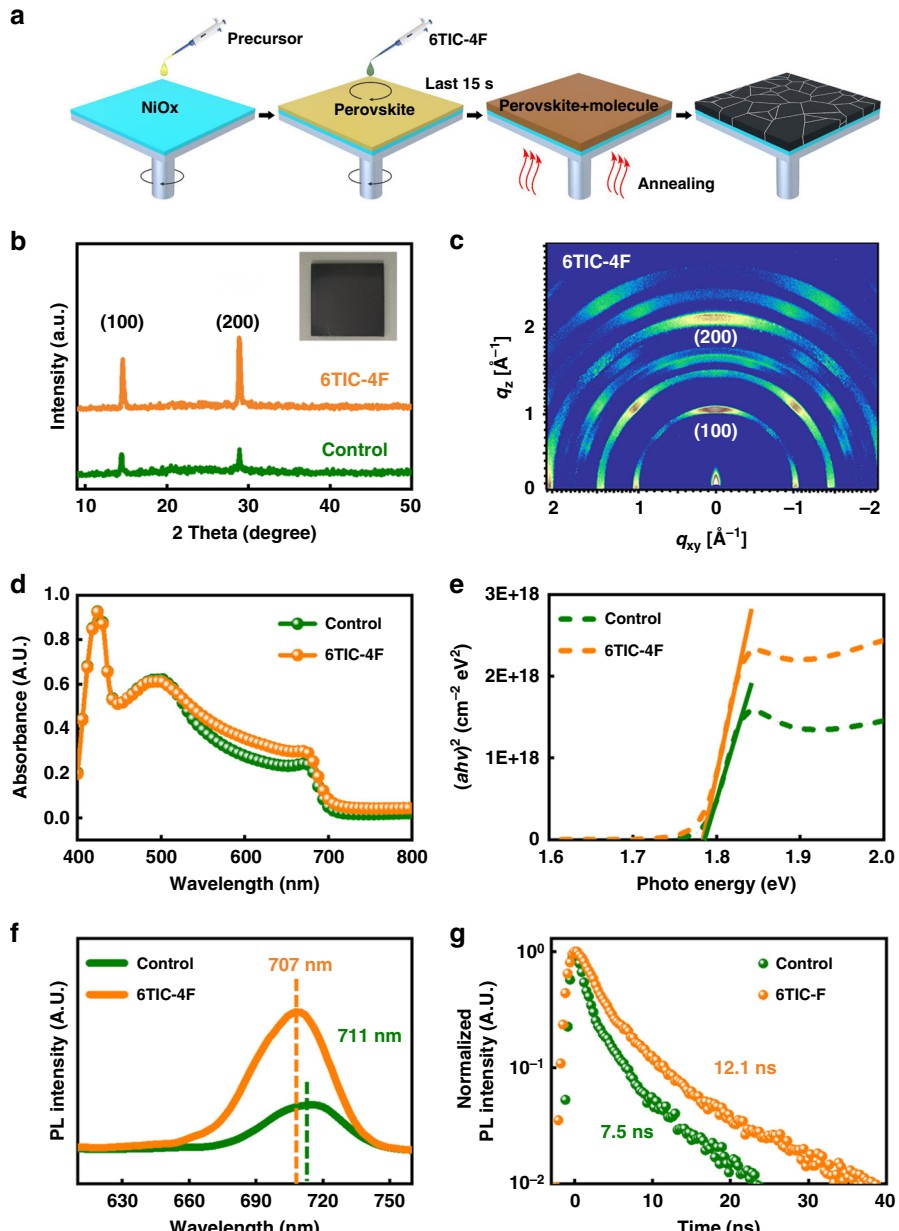

**Fig. 1 Film fabrication and characterization. a** Illustration of the deposition processes of the prepared CsPbBr$_x$I$_{3-x}$ film in this study. **b** X-ray diffraction patterns, (inset: the photograph of CsPbBr$_x$I$_{3-x}$ film with 6TIC-4F treatment), **c** grazing incidence wide-angle X-ray scattering, and **d**, **e** the UV–vis absorption spectra and Tauc plot of the prepared CsPbI$_x$Br$_{3-x}$ films without (green) and with (orange) 6TIC-4F passivation. **f** Steady-state photoluminescence (SSPL) spectra and **g** normalized time-resolved photoluminescence (TRPL) decay profiles of the studied CsPbBr$_x$I$_{3-x}$ films without (green) and with (orange) 6TIC-4F passivation on quartz substrates.

results provided in Fig. 1c and Supplementary Fig. 2 also showed that films with 6TIC-4F treatment have the diffraction patterns of perovskite remained with brighter spots, confirming its better crystallinity without structural changes. We then carried out scanning electron microscopy (SEM) to investigate the effects of 6TIC-4F treatment on the perovskite surface morphology. As seen, both perovskite films showed homogenous surface morphology; however, the average grain size of the film increased from 110 nm (Supplementary Fig. 3a) to 200 nm after 6TIC-4F treatment (Supplementary Fig. 3b). It has been reported that the intermediate solvates (perovskite solvent) formed during spin-coating process and could impact the morphology of perovskite film[38,39]. The nucleation and growth of the solvates can be promoted by introducing small molecules with Lewis base

functional groups[29]. These large crystalline solvates could act as a template for perovskite growth when the solvent was removed, leading to the formation of perovskite phase. Therefore, we speculate that 6TIC-4F could facilitate the nucleation and growth of intermediate solvates when it was introduced into perovskite precursor solution during spin-coating process, and then these solvates would act as templates for further growth of perovskites, resulting in larger grain size and denser morphology than those of the control film.

Presented in Fig. 1d is the UV–vis absorbance of both films with a thickness of about 350 nm. The thicknesses of films were measured by using a DektakXT Profiler (Bruker). As seen, the film's absorbance was enhanced after 6TIC-4F treatment. The bandgap of both films was ~1.78 eV as extracted from the Tauc

plot (Fig. 1e), suggesting the negligible influence of 6TIC-4F treatment on resultant electronic structures of perovskite. Afterward, steady-state photoluminescence and time-resolved photoluminescence (TRPL) were conducted to investigate the radiative recombination behavior and charge carrier dynamics in the $CsPbI_xBr_{3-x}$ films without and with 6TIC-4F treatment. As indicated in Fig. 1f, the PL peak of film after 6TIC-4F treatment blue shifted from 711 to 707 nm, possibly as the consequence of reduced shallow defects on the $CsPbI_xBr_{3-x}$ grain boundaries and surfaces[40,41]. Besides, the PL intensity increased obviously, implying that non-radiative recombination within the $CsPbI_xBr_{3-x}$ film was significantly suppressed. The charge carrier lifetime of the perovskite films were obtained by fitting the TRPL spectra to single-exponential function. As illustrated in Fig. 1g, the PL lifetime of the $CsPbBr_xI_{3-x}$ film was increased from 7.5 to 12.1 ns after the 6TIC-4F treatment, confirming the reduced possibility of defect-induced recombination. All these results clearly unveil the defect passivation function of the 6TIC-4F treatment.

**Device performance**. The effects of 6TIC-4F passivation on device performance were then evaluated by fabricating an inverted device with a structure of ITO/NiO$_x$/$CsPbI_xBr_{3-x}$/ZnO/C60/Ag (Fig. 2a). As shown in Fig. 2b and Table 1, the control device derived from the pristine film delivered a modest PCE of 13.9% with a $V_{OC}$ of 1.10 V, a short-circuit current density ($J_{SC}$) of 17.0 mA cm$^{-2}$, and a fill factor (FF) of 74.2%. Whereas, the champion device based on the 6TIC-4F-treated film exhibited an encouraging PCE of 16.1% with a higher $V_{OC}$ of 1.16 V, a $J_{SC}$ of 17.7 mA cm$^{-2}$, and an FF of 78.6%. The calculated $J_{SC}$ (Table 1) based on the external quantum efficiency (EQE) plots in Fig. 2c were close to the values obtained from current density–voltage ($J$–$V$) measurements, confirming the reliability of the $J$–$V$ results. Note that the device performance was tested under both reverse and forward scans, and the $J$–$V$ curves showed negligible hysteresis (Supplementary Fig. 4a, b). The histograms of PCE from 32 devices based on the films without and with 6TIC-4F passivation are presented in Supplementary Fig. 5, proving the high reproducibility of our fabricated devices.

The stabilized power output of devices was recorded with tracking the photocurrent at the maximum power point in a nitrogen-filled glovebox. As shown in Fig. 2e, the photocurrent of the control device gradually declined that might be due to the presence of significant trap states in the $CsPbI_xBr_{3-x}$ film. Whereas, the PVSC based on the 6TIC-4F-treated film showed a highly stable photocurrent with a negligible loss (15.6 mA cm$^{-2}$) after 500 s. In order to carefully evaluate the stabilized device performance, the champion device was sent to the National Institute of Metrology, Beijing, China (NIM) for certification, and a certified PCE of 15.6% was obtained (please refer to the details in Table 1 and Supplementary Fig. 6). Besides, the photostability of these fabricated PVSCs was also characterized under continuous one sun equivalent illumination under a $N_2$ atmosphere. It needs to be pointed out that the interval measurements of device performance were conducted under continuous one sun equivalent illumination for 350 h in our photostability test rather than keeping devices under continuous working condition. Figure 2f clearly illustrated that the 6TIC-4F-treated PVSC can retain roughly 85% of initial PCE after light soaking for 350 h; in contrast, the control device degraded to only 65% of its original PCE. It is known that the charged defects on perovskite surface and/or grain boundaries have lower reaction active energy, which more easily results in the degradation of perovskite film under the attacks of moisture, oxygen, or light[42]. Therefore, the 6TIC-4F passivation should prevent the defects from these attacks. In addition, as shown in Supplementary Fig. 13, the contact angle of

$CsPbI_xBr_{3-x}$ film surface increased from 34.6° to 40.1° after 6TIC-4F treatment, implying a more hydrophobic film with 6TIC-4F. These results showed that 6TIC-4F passivation should improve the stability of $CsPbI_xBr_{3-x}$-based PSCs because of the enhanced resistance for moisture, oxygen, or light. This clearly shows the advantages of the 6TIC-4F-treated film. In addition, it is noteworthy that the phase segregation in $CsPbI_xBr_{3-x}$ films will be significantly alleviated when the value of x is larger than 2.01[22]. Herein, the I/Br stoichiometry ratio in precursor solution (Materials and solution preparation section in Methods section) is 14, which means the x value is 2.8. Therefore, the phase segregation in our $CsPbI_xBr_{3-x}$ films should also be suppressed like literature reported. In addition, the shape of PL spectra of $CsPbI_xBr_{3-x}$ films with and without 6TIC-4F treatment only changed slightly under continuous one sun equivalent illumination for 30 min (Supplementary Fig. 14), demonstrating the good phase stability of our $CsPbI_xBr_{3-x}$ films. Furthermore, the surface defects have been shown to catalyze phase segregation in mixed-ion halide perovskites due to carriers trapping and charge accumulations at perovskite surface[43]. This indicates that the photostability of our $CsPbI_xBr_{3-x}$ films should be enhanced after 6TIC-4F passivation.

As seen, the improved PCE observed in the 6TIC-4F-treated device is mainly attributed to the improved $V_{OC}$ and FF, and the possible underlying mechanisms will be discussed later. It is worth noting that the LUMO level of 6TIC-4F is about −4.14 eV[33], which just sits between the CBM of perovskite (−3.49 eV, Supplementary Fig. 7) and ZnO (−4.25 eV)[34]. Electrons can then be extracted from the $CsPbI_xBr_{3-x}$ film into the 6TIC-4F molecules. Due to the tendency for 6TIC-4F to form good packing[33], it will facilitate electron transport and collection for 6TIC-4F-treated $CsPbI_xBr_{3-x}$ PVSC to result in higher $J_{SC}$ and FF. This will enable a better energy alignment at the perovskite/ZnO interface to facilitate the charge transfer across this interface (Fig. 2d).

In addition, the $CsPbI_xBr_{3-x}$ film showed larger grain size and a denser morphology after the 6TIC-4F treatment (Supplementary Fig. 3), which could result in fewer defects on grain boundaries[44,45] and less current leakage in devices[46,47]. In order to distinguish the effect of morphology on the improved performance, the control devices with 6TIC-4F posttreatment (Bilayer 6TIC-4F) were fabricated, and compared with the control devices and devices based on 6TIC-4F-containing antisolvent-treated films. The performance of the champion devices is listed in Supplementary Fig. 16 and Supplementary Table 2. The $V_{OC}$, $J_{SC}$, and FF of the control device with bilayer 6TIC-4F increased to 1.14 V, 17.1 mA cm$^{-2}$, and 76.5%, respectively. Comparing the control device and device treated with 6TIC-4F containing antisolvent, it is clear that the major contribution from improved morphology is the increased $J_{SC}$, which may derive from the denser film. Therefore, the improved $V_{OC}$ and FF may mainly come from the passivation effect of 6TIC-4F on $CsPbI_xBr_{3-x}$ films.

**The potential passivation mechanism**. In order to verify the potential interactions between $CsPbI_xBr_{3-x}$ and 6TIC-4F, X-ray photoelectron spectroscopy (XPS) measurements were conducted to characterize the elemental states on the $CsPbI_xBr_{3-x}$ surface before and after the 6TIC-4F treatment. Supplementary Fig. 12 presented the states of Pb of the $CsPbI_xBr_{3-x}$ film and showed two feature peaks of Pb $4f_{7/2}$ and Pb $4f_{5/2}$. The Pb $4f_{7/2}$ core level can be deconvoluted into two peaks located at 136.8 eV and 138.1 eV, respectively. The intensity of $Pb^{2+}$ (138.1 eV) was significantly higher than that of $Pb^0$ (136.8 eV) and indicated the existence of $Pb^{2+}$ defects. As shown in Supplementary Fig. 15, the XPS feature peaks of Pb $4f$ and N $1s$, S $2p$ showed obvious shift

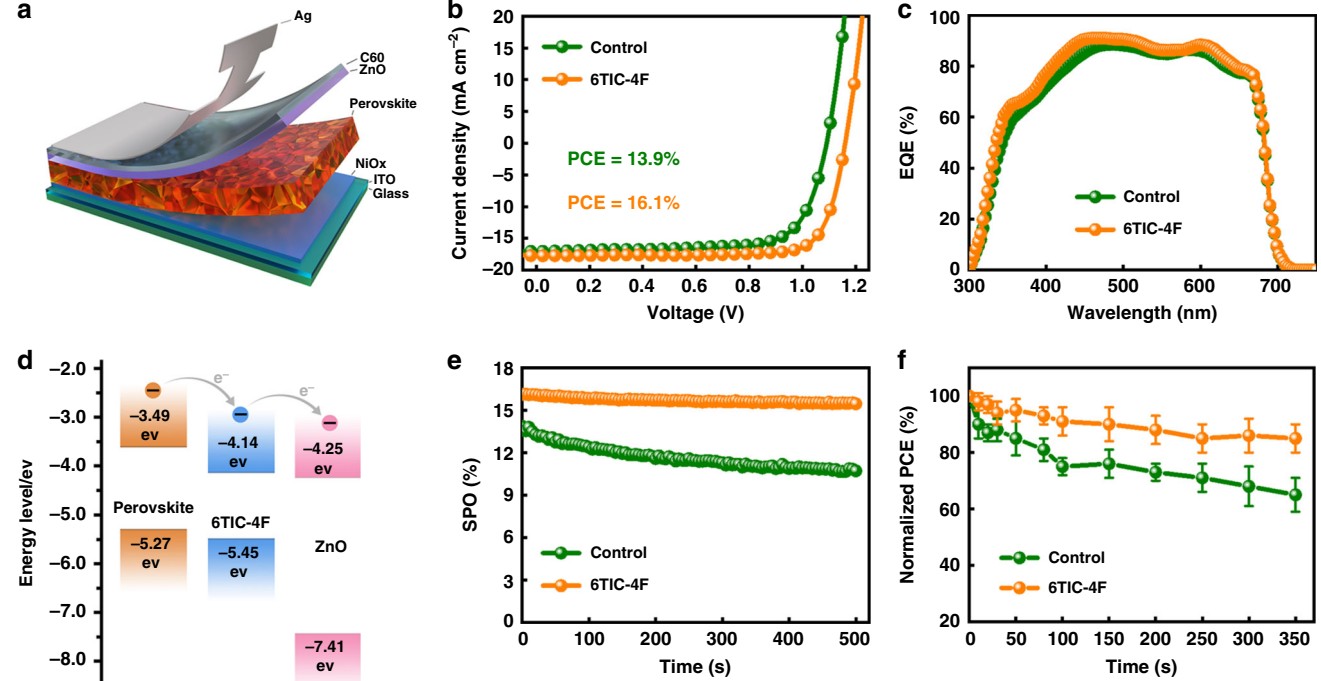

**Fig. 2 Device performance. a** Device architecture and **d** the energy-level diagram of the studied inverted CsPbI$_x$Br$_{3-x}$ PVSC. **b** The J–V curves and **c** the EQE spectra of the champion CsPbI$_x$Br$_{3-x}$ PVSCs without (green) and with (orange) 6TIC-4F passivation. **e** Stabilized power output tests and **f** normalized efficiencies of the studied CsPbI$_x$Br$_{3-x}$ PVSCs under continuous one sun equivalent illumination. The error bars represents the standard deviations for PCE of the devices.

**Table 1 Photovoltaic parameters of CsPbI$_x$Br$_{3-x}$ PVSCs without and with 6TIC-4F passivation under AM (air mass) 1.5G illumination.**

| Sample | $V_{OC}$ [V] | $J_{SC}$ [mA cm$^{-2}$] | FF [%] | PCE [%] |
|---|---|---|---|---|
| Control | 1.10 | 17.00 (16.00) | 74.20 | 13.90 |
| 6TIC-4F | 1.16 | 17.70 (16.50) | 78.60 | 16.10 |
| NIM certificated cell | 1.145 | 17.44 | 78.00 | 15.60 |

The currents in brackets are the EQE-integrated $J_{SC}$

towards smaller and larger binding energy, respectively, after 6TIC-4F treatment, while the feature peaks of Cs 3$d$, F 1$s$, and O 1$s$ only shifted slightly. These XPS results indicated the potential interaction between Pb on CsPbI$_x$Br$_{3-x}$ and -CN/-S groups on 6TIC-4F. Then, density functional theory (DFT) calculations were conducted on pristine Cs$_{56}$Pb$_{27}$I$_{108}$ cluster (Supplementary Fig. 9) and Pb exposed Cs$_{44}$Pb$_{27}$I$_{99}$ cluster (Fig. 3b). To model the surface passivation effect of the 6TIC-4F molecule, we have sampled several docking positions of 6TIC-4F on the uncompensated Pb surface, allowing full relaxation of the 6TIC-4F molecules, and then carried out the electronic structure computation. The 6TIC-4F surface density is about 0.003 Å$^{-2}$. The optimized structures are shown in Supplementary Fig. 8. It can be observed that F-Pb, N-Pb, and S-Pb bonds stably formed after the geometry optimizations. The formation energies of the five motifs are −2.90 eV, −2.81 eV, −3.14 eV, −2.20 eV, and −3.61 eV, respectively, for Supplementary Fig. 8a–e. It is noteworthy that the S-Pb bonds can only be formed when the 6TIC-4F is on the edges of the grain boundaries due to the strong steric effect of the side chains. Therefore, after excluding the S-Pb bonding motif, the N-Pb bonding motif with a formation energy of −3.14 eV is going to have an over 99% Boltzmann distribution ratio, which is

likely to be the most effective electron-donating group to passivate most of the surface traps. This simulation is consistent with the XPS results that only the interaction between Pb and -CN/-S were detected. Notably, when the 6TIC-4F molecule is removed, the electron density was localized on the Pb-exposed surface, which is a typical indication for the surface traps (Fig. 3b). However, Fig. 3e clearly shows that the valence electron densities of the most stable 6TIC-4F/Cs$_{44}$Pb$_{27}$I$_{99}$ motif (corresponding to Fig. 3a) shifts toward the bulk of the cluster and approaches the scenario as depicted in Supplementary Fig. 9. These calculation results thus clearly indicate the elimination of surface traps after the 6TIC-4F passivation (Fig. 3d, f).

We then employed the space-charge-limited current method to confirm this result. The J–V curves of the electron-only devices with structure ITO/SnO$_2$/CsPbI$_x$Br$_{3-x}$/ZnO/Ag are plotted in Fig. 3c, where the kink point voltage is determined as the trap-filling limited voltage ($V_{TFL}$) that is closely related to defect density ($N$)[48,49]. The measured $V_{TFL}$ for the pristine CsPbI$_x$Br$_{3-x}$ film is ~1.08 V while the value is decreased to 0.87 V after 6TIC-4F passivation. Accordingly, the calculated $N$ for the films without and with 6TIC-4F passivation is $3.41 \times 10^{16}$ cm$^{-3}$ and $2.75 \times 10^{16}$ cm$^{-3}$, respectively. The reduced $N$ observed in the 6TIC-4F-treated film clearly manifests the defect passivation function of 6TIC-4F, which efficiently suppresses the non-radiative recombination in the derived device.

**Device physics and recombination process.** As mentioned earlier, the performance enhancement is mainly due to the improved $V_{OC}$ and FF. To better understand the underlying details of these improvements, a series of device analyses were carefully performed. First, the $V_{OC}$ of PVSCs was measured under a range of light intensities ($P_{light}$) and plotted as a function of $P_{light}$ in logarithm scales as showed in Fig. 4a. In principle, the slope of this plot is correlated with the ideality factor ($n$) in the form of

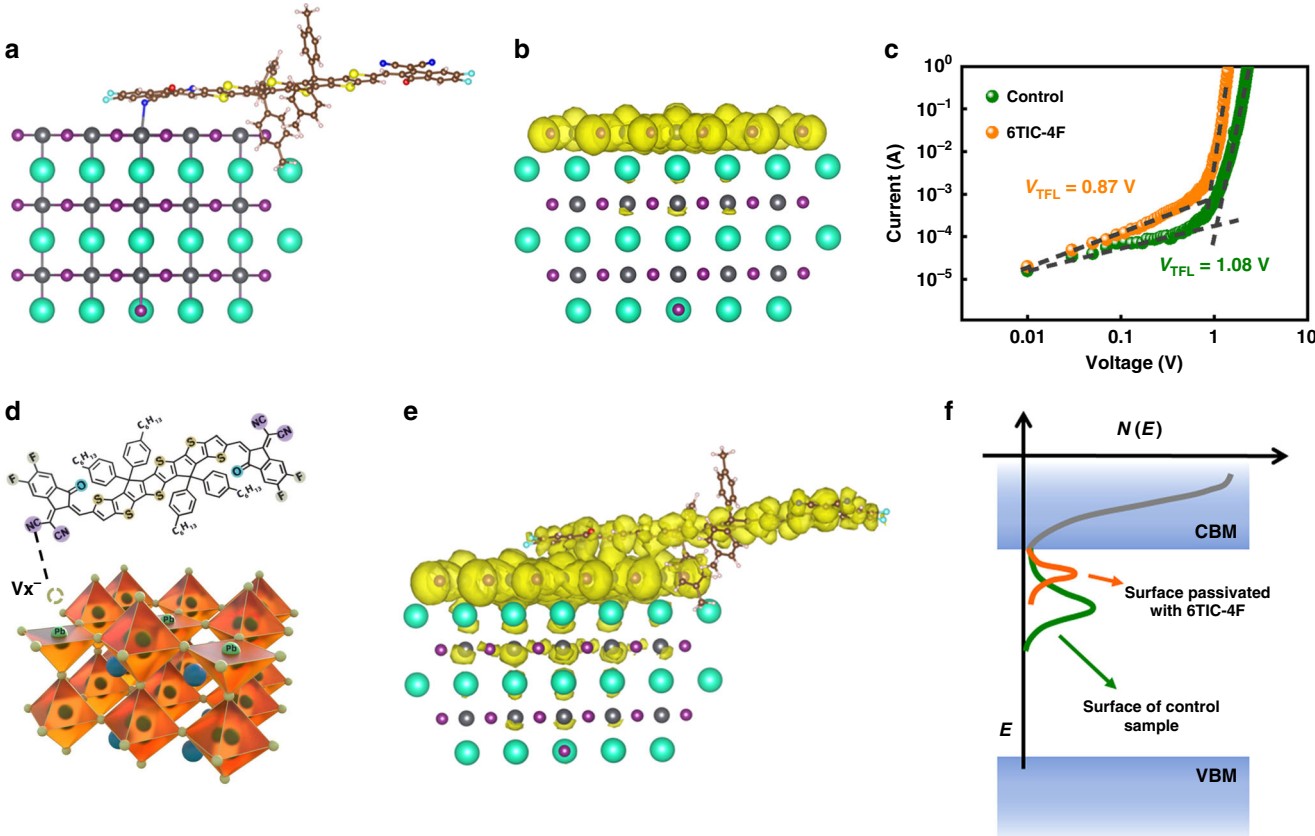

**Fig. 3 Theoretical simulation. a** Illustration of the perovskite and 6TIC-4F structures for DFT calculation. **b** The valence electron density of the Pb exposed to $Cs_{44}Pb_{27}I_{99}$ cluster. **c** J–V characteristics of devices with $ITO/SnO_2/CsPbI_xBr_{3-x}/ZnO/Ag$ configuration for estimating the defect density in the films. **d** Illustration of possible passivation mechanism and potential interaction sites. **e** The valence electron density of the most favorable $6TIC-4F/Cs_{44}Pb_{27}I_{99}$ motif. **f** The energy-level diagram of the trap states passivation by 6TIC-4F.

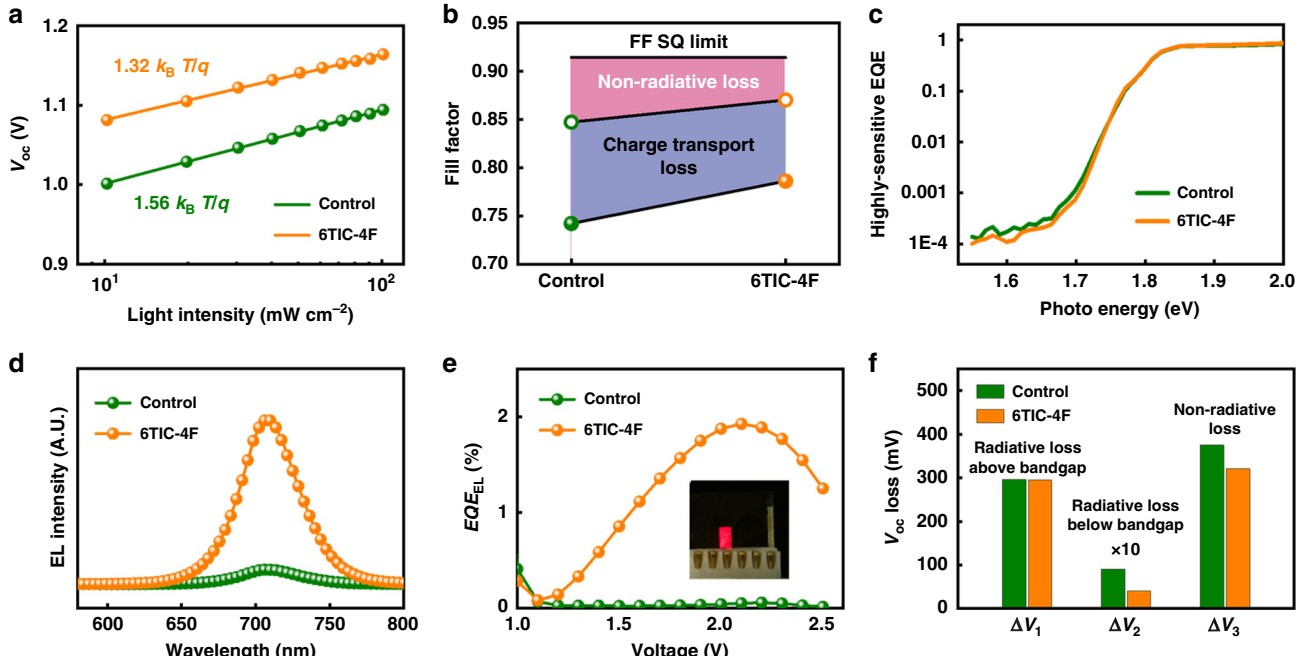

**Fig. 4 Device physics and recombination process. a** The plot of light intensity dependent $V_{OC}$ of the studied devices. **b** The device FF limitation is composed of non-radiative loss (pink area) and charge transport loss (blue area). The solid and open circles stand for the measured FF and the maximum FF without charge transport loss, respectively. **c** The highly sensitive EQE of the studied PVSCs. **d** EL spectra of the studied PVSCs operating as LEDs. **e** EQE of EL of the PVSCs working in LEDs mode under different voltage, inset: LED working image. **f** The radiative and non-radiative $V_{OC}$ loss of the studied PVSCs.

$nk_BT/q$, which is the reflex of trap-induced recombination behavior in PVSCs[50,51]. Compared with the control device ($n = 1.56$), the 6TIC-4F-treated PVSC showed a smaller $n$ of 1.32, indicating the suppressed trap-induced recombination in the device. Furthermore, the FF loss between the Shockley–Queisser limit and measured FF is composed of non-radiative loss and charge transport loss, and the maximum FF ($FF_{max}$) without charge transport loss can be empirically calculated with the equation $FF_{max} = \frac{\nu_{OC} - \ln(\nu_{OC} + 0.72)}{\nu_{OC} + 1}$, where $\nu_{OC} = \frac{V_{OC}}{nk_BT/q}$[52]. As illustrated in Fig. 3, the -CN groups on the 6TIC-4F can passivate the surface traps of perovskite film and reduce the non-radiative loss. Meanwhile, the improved charge transfer in $CsPbI_xBr_{3-x}$ film with 6TIC-4F treatment benefits the charge collection, which has been discussed above, and thus reduce the charge transport loss. As shown in Fig. 4b, the non-radiative loss in the 6TIC-4F-treated device was clearly suppressed, while the charge transport loss was also slightly decreased. These results verified that 6TIC-4F passivation not only suppresses the non-radiative recombination in the device, but also improves charge transport in the device.

Both transient photovoltage (TPV, Supplementary Fig. 10a) and transient photocurrent (TPC, Supplementary Fig. 10b) measurements were also carried out to investigate the device recombination dynamics. By fitting the TPV and TPC curves to a biexponential decay function, the 6TIC-4F-treated PVSC showed a charge-recombination lifetime of 4.7 μs and charge-extraction time of 0.54 μs, whereas the control device showed a lifetime of 3.1 μs and 0.79 μs, respectively. These results again affirms that the 6TIC-4F passivation efficiently suppresses charge recombination and facilitates the charge extraction efficiency, being consistent with the above FF analysis.

According to the detailed balance theory[53–55], the $V_{OC}$ loss can be attributed to three factors:

$$q\Delta V = E_g - qV_{OC}$$
$$= \left(E_g - qV_{OC}^{SC}\right) + \left(qV_{OC}^{SC} - qV_{OC}^{rad}\right) + \left(qV_{OC}^{rad} - qV_{OC}\right)$$
$$= \left(E_g - qV_{OC}^{SC} + q\Delta V_{OC}^{SC}\right) + \Delta qV_{OC}^{rad} + \Delta qV_{OC}^{non-rad}$$
$$= q(\Delta V_1 + \Delta V_2 + \Delta V_3)$$
$$(1)$$

where $q$ is the elementary charge, $\Delta V$ is the total voltage loss, $E_g$ is the bandgap of perovskite, $V_{OC}^{SC}$ is the Shockley–Queisser limit, $V_{OC}^{rad}$ is the $V_{OC}$ when only radiative recombination occurred in PVSCs, $\Delta V_{OC}^{SC}$ is the $V_{OC}$ loss due to the non-ideal EQE above bandgap, $\Delta V_{OC}^{rad}$ is the $V_{OC}$ loss of the sub-bandgap radiative recombination, and $\Delta V_{OC}^{non-rad}$ is the $V_{OC}$ loss of non-radiative recombination. More details of $V_{OC}$ loss calculation can be found in Supplementary Note 1.

Three terms of $V_{OC}$ loss ($\Delta V_1$, $\Delta V_2$, and $\Delta V_3$) were showed in Fig. 4f and summarized in Table 2. In principle, $\Delta V_1$ is due to the radiative recombination above $E_g$, which is unavoidable in a solar cell device, and the non-ideal EQE above $E_g$. The PVSCs without and with 6TIC-4F passivation had similar $\Delta V_1$ of 296.16 mV and 295.36 mV, respectively. On the other hand, $\Delta V_2$ comes from the energy loss associated with extra thermal radiation in a solar cell device in dark, where EQE of PVSCs extends into the region

below $E_g$ and induces more black-body radiation. The highly sensitive EQE of these devices was then measured to calculate $\Delta V_2$ (Fig. 4c)[56–58], and the calculated value is 11.32 mV for the control device and 6.88 mV for the 6TIC-4F-teated device, respectively. The small $\Delta V_2$ observed in the treated device could be ascribed to the steep absorption edge of our perovskite and consistent with the previous reports[59–61]. Finally, $\Delta V_3$ is the $V_{OC}$ loss due to the non-radiative recombination, which can be evaluated with the function of $\frac{k_BT}{q}\ln(EQE_{EL})$. The $k_B$ and $EQE_{EL}$ are the Boltzmann constant and EQE of electroluminescence (EL), respectively. Hence, the PVSCs were operated in a light-emitting diode (LED) mode to acquire the associated $EQE_{EL}$[27,62,63]. As shown in Fig. 4d, e, the 6TIC-4F-treated PVSC presented much a higher EL intensity and $EQE_{EL}$ than that of the control PVSC. The calculated $\Delta V_3$ of the 6TIC-4F-treated PVSC is about 317.76 mV, which is lower than the value of the control device by 54.76 mV. As discussed earlier, the $V_{OC}$ improvement enabled by the 6TIC-4F passivation is mainly ascribed to the suppressed non-radiative recombination, which increases the $EQE_{EL}$ by almost one order of magnitude as observed.

## Discussions

In summary, our work has demonstrated an effective passivation strategy by using a Lewis base, 6TIC-4F, to achieve a high-performance inverted inorganic PVSC that can deliver a champion PCE of 16.1% and a certified PCE of 15.6% with improved photostability, representing the most efficient inverted inorganic PVSCs to date. Our DFT calculations revealed the potential passivation mechanism between 6TIC-4F and the $CsPbI_xBr_{3-x}$ film: the electron-rich CN group in 6TIC-4F directly interacts with the exposed Pb on perovskite surface through coordination bonds to passivate the perovskite trap states and delocalize the valence electron density from the Pb-exposed surface to bulk, which is beneficial for suppressing the non-radiative recombination. The detailed analysis of $V_{OC}$ and FF also demonstrated that the enhancements were mainly resulted from the reduced non-radiative recombination in the device, enabled by the 6TIC-4F passivation. The $EQE_{EL}$ of the 6TIC-4F-treated device can be largely increased by almost one order of magnitude. In addition, the appropriate LUMO of 6TIC-4F enables the better energy-level alignment between CBM of perovskite and ZnO, thereby improving the charge extraction from perovskite to the charge transport layer. This work provides insights of mechanism for surface trap passivation of inorganic perovskites, which signifies the importance of rational design of functional interlayers for all-inorganic PVSCs to realize improved performance and stability.

## Methods

**Materials and solution preparation.** 6TIC-4F was synthesized in our own lab according to our published paper[64]. NiO$_x$ nanoparticles (NPs) and ZnO NPs were synthesized according to previously reported procedures[65,66]. Cesium iodide (CsI, 99.99%) and Cesium bromide (CsBr, 99.999%) were purchased from Xi'an Polymer Light Technology Corp. and Sigma-Aldrich, respectively. SnO$_2$ colloid (tin (IV) oxide) precursor was purchased from Alfa Aesar. C60 was purchased from Nano C. Dimethylformamide (DMF, > 99.0%) and dimethyl sulfoxide (DMSO) were both purchased from TCI. Unless otherwise stated, all of the chemicals were obtained commercially and used directly without purification. The perovskite precursor solution was prepared by dissolving 1.0 mmol PbI$_2$, 0.2 mmol CsBr and 0.8 mmol

**Table 2 The $V_{OC}$ loss analysis of the studied devices without and with 6TIC-4F passivation.**

| Sample | $E_g$ [eV] | $V_{OC}^{SC}$ [V] | $V_{OC}$ [V] | $\Delta V_1$ [mV] | $\Delta V_2$ [mV] | $\Delta V_3$ [mV] |
|---|---|---|---|---|---|---|
| Control | 1.78 | 1.49 | 1.10 | 296.16 | 11.32 | 372.52 |
| 6TIC-4F | 1.78 | 1.49 | 1.16 | 295.36 | 6.88 | 317.76 |

CsI in a mixture solvent of DMSO and DMF (9:1 v/v, 1 mL), and then kept stirring overnight in an $N_2$-filled glovebox.

**Device fabrication and characterization**. The ITO glasses were cleaned with detergent, DI water, acetone, and isopropanol sequentially by sonicating for 10 min. Then, the ITO glasses were dried in an oven with 100 °C. The ITO glass was treated with oxygen plasma for 3 min before using. The $NiO_x$ NPs solution was spin-coated onto the treated ITO with a speed of 3000 rpm, and then the as-prepared $ITO/NiO_x$ substrates were moved into an $N_2$-filled glovebox. The perovskite active layers were prepared onto the $ITO/NiO_x$ substrates by a spin-coating procedure composed of 1500 rpm for 10 s and 5000 rpm for 30 s. When the spin-coating speed reached 5000 rpm, 0.2 mL CB or 6TIC-4F in CB (3 mg mL$^{-1}$) was dripped onto the precursor for control film or film with 6TIC-4F, respectively, at 15 s before the end of processing. The films were then annealed at 55 °C for 50 s followed by 255 °C for 50 s to obtain black-phase perovskite. ZnO ETL was prepared by spin-coating 50 μL ZnO NPs solution onto perovskite film. Finally, 10 nm of $C_{60}$ and 90 nm of silver were thermally evaporated ($2 \times 10^{-6}$ mbar) onto the top layer of samples through a shadow mask to finish the device preparation.

$J$–$V$ characteristics were implemented with a solar simulator (Enlitech, SS-F5, Taiwan) under AM 1.5 G illumination in $N_2$-filled glovebox at room temperature. The AM 1.5 G solar simulator's light intensity was identified with a National Renewable Energy Laboratory calibrated silicon solar cell with a KG5 filter. The $J$–$V$ curves under forward and reverse scan were both collected with a scan rate of 0.1 V s$^{-1}$. A mask with aperture area of 0.0672 cm$^2$ was used during $J$–$V$ measurement. EQE result was collected by Enlitech QE-3011 system. Zeiss EVO 18 SEM was undertaken to analysis the morphologies of films. The obtained PL spectrum was from FLS920 spectrofluorometer (Edinburgh) and an HP 8453 spectrophotometer. A digital oscilloscope (Tektronix TDS 3052 C) was used to record TPC and TPV of the PVSCs. The thicknesses of films were measured by using a DektakXT Profiler (Bruker).

**DFT calculation details**. A $Cs_{56}Pb_{27}I_{108}$ cluster has been extracted from the optimized $\alpha$-$CsPbI_3$ supercell. In the cluster model, all Pb atoms are fully coordinated with the I atoms in the octahedron manner. All the cluster calculations are performed in gas phase in Gaussian 16 software package[67]. The geometry optimizations and electron density calculations are carried out using the PBE0 hybrid density functional with Los-Alamos double-$\zeta$ pseudopotential basis set[68]. When later optimizing the 6TIC-4F docking positions, the Grimme's DFT-D3 corrections has been used to consider the van der Waals interaction[69].

**Reporting summary**. Further information on research design is available in the Nature Research Reporting Summary linked to this article.

## Data availability

The authors declare that the main data supporting the findings of this study are available within the article and its Supplementary Information files. Extra data are available from the corresponding author upon reasonable request.

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

## Acknowledgements

This work was partially supported by the National Science Foundation (DMR-1608279), the Office of Naval Research (N00014-17-1-2260), and the Air Force Office of Scientific Research (FA9550-18-1-0046). A.J. and Z.Z. thank the CityU start-up fund (7200587, 9610421) and Innovation and Technology Fund (ITS/497/18FP, GHP/021/18SZ). Z.Z. will also thank Natural Science Foundation of Guangdong Province, (2019A1515010761). We thank M. Li for the help in highly-sensitive EQE measurement. Theoretical research is supported by the National Science Foundation (CHE-1856210 to X.L.). This work was facilitated though the use of advanced computational, storage, and networking infrastructure provided by the Hyak supercomputer system and was funded by the STF at the University of Washington and the National Science Foundation (MRI- 1624430).

## Author contributions

J.W. and J.Z. contributed equally to this work. J.W. and J.Z. designed all experimental investigations and conducted devices characterization. J.W. fabricated devices. J.Z. analyzed detailed $V_{OC}$ loss. Y.Z., Q.X. and C.C. assisted with experiments and data analysis. H.L. and X.L. conducted DFT calculation. H.Y., Z.Z. and A.K.Y.J. oversaw the project, and contributed to all aspects of analysis and experimental design. J.Z. and J.W. wrote the manuscript with assistance from H.Y., Z.Z. and A.K.Y.J.

## Competing interests

The authors declare no competing interests.
