## [Peer Review File · Nature Communications]

Reviewers' comments:

Reviewer #1 (Remarks to the Author):

The manuscript by Wang et al. reports high efficiency all-inorganic perovskite solar cells enabled by defect passivation using a Lewis base 6TIC-4F. The passivation effect of 6TIC-4F is proved by the enhanced PL lifetime, improved Voc and EQE(EL) as well as reduced defect density calculated from the trap-filling limited voltage. The authors also tried to understand the passivation mechanism by theoretical simulation. The device performance of the solar cells based on the 6TIC-4F-treated perovskite film is indeed impressive compared to those of the reported inverted all-inorganic perovskite solar cells. Some questions before the publication of this ms:

1. In the title, "luminance" might not be the right word. Luminance is a measure of light output for visible LEDs with an unit of cd/m². However, there is no data showing the luminance of the solar cells in this manuscript.
2. 6TIC-4F has many electron-rich functional groups such as -F, -C=O, -S and even the fused ring (Adv. Mater. 2018, 30, 1707583) can act as the passivation unit for perovskites. Experimental evidences (e.g., FTIR, XPS, or Raman) should be provided to further understand the passivation mechanism.
3. More details are suggested with regard to the photostability test. Are the devices kept working under light illumination during the photostability test? In addition, could the authors explain why the mixed halide perovskites used in this work are stable? There are many papers reporting phase segregation of mixed halide perovskites under light illumination (e.g., ACS Energy Lett. 2018, 31, 204-213; Chem. Sci., 2015, 6, 613-617; Nature Communications 2017, 8, 200).
4. 6TIC-4F is introduced into the perovskite films by anti-solvent. Does 6TIC-4F function as well if it is introduced through perovskite precursor solution or post-treatment (e.g., spin-coating 6TIC-4F solution on top of annealed perovskite films)?
5. In addition to the passivation effect, the enhanced film quality by 6TIC-4F could be another reason for the enhanced device performance. What is the mechanism of the enhanced grain size after 6TIC-4F treatment?

Reviewer #2 (Remarks to the Author):

In this work, Wang et. al present a defect passivation strategy for improving the performance of a Cs-based halide perovskite in the p-i-n device architecture. The study has been carefully conducted and the improvements in the resultant films and devices are impressive. There are, however, a few minor comments that I believe should be addressed before this paper is published.

- 1) The grammar and overall language in this article needs to be carefully checked for errors, as there are many of them which makes reading the paper a bit difficult at times. Also, redundancies such as "photo-illumination" need to be corrected.
- 2) This paper is not very well referenced, particularly in the introduction. The introduction should be framed such that the work in this paper can be seen in proper context of the work that has already been done. The main focus of this work is passivation of the surfaces and grain boundaries with 6TIC-

4F, and there is a large body of work that deals with the passivation of perovskite surfaces: lewis base passivation, passivation, as well as passivation using quarternary ammonium cations in both n-i-p and p-i-n solar cells. Work that has been done in this vein should at least be mentioned/referenced if not discussed. Additionally, for discussing the instability of dopants, the authors might consider citing the following papers (10.1002/aenm.201800554, 10.1039/C8TC02242A). The papers which the authors have referenced speak to the instability of Li-doped spiro but do not mention the problems with tbp.

Reviewer #3 (Remarks to the Author):

In this manuscript, the authors reported a small organic molecule 6TIC-4F which assists crystallization of perovskite as well as charge transfer via suitable band alignment. The PCE of p-i-n structured PSCs has achieved 16.1% and a certified value of 15.6%. This manuscript may be accepted by Nature Communications after the authors clearly address the following concerns.

1. Considering the multiple functional "-CN" groups, the large molecular size of 6TIC-4F and obvious enhancement of PV performance, it is important to provide evidence on how 6TIC-4F is distributed in the perovskite film and add relevant discussions in the maintext. How the charge carriers behavior at the grain boundaries and/or film surface based on 6TIC-4F distribution pattern? Does steric effect of 6TIC-4F also need to be taken into consideration in terms of "passivation"? One of the -CN groups in Figure S1a is missing. Please check.
2. Page 5 line 133, "by dropping 130 mL solution of 6TIC-4F in chlorobenzene", should be μL . In addition, the authors describe the "Device Fabrication" procedure in the supporting information as "the perovskite active layer was prepared by a widely used two-step spin-coating procedure at 1500 rpm (for 10 s) and 5000 rpm (for 30 s). When the spin-coating reached 5000 rpm, 0.2 ml 6TIC-4F (3 mg/ml, CB) was dripped onto the substrates." Please be consistent with those expressions in the maintext, e. g. one-step or two-step? 0.2 ml or 130 μL ? ml or mL? Fabrication of the control film should also be provided.
3. Page 6, line 135, the authors state that "followed by annealing at 255 °C for another 50 s to get the black phase perovskite", please provide TGA measurement for 6TIC-4F both of its pure form and with perovskite.
4. Please provide film thickness when comparing the absorbance in Figure 1d.
5. Characterizations such as XPS should be provided to show the existence of Pb^{2+} or metallic Pb defects. Apparently the authors only discussed the under coordinated Pb^{2+} type of defect, does the film morphology also impacts the PV performance, and why? Please add discussion.
6. It would be fair to prepare control film with high quality (pinhole free) to make comparison rather than that shown in Figure S3a as there are already many mature methods with which such films can be achieved.
7. Please give reasons for enhancing the stability of PSCs after 6TIC-4F treatment.
8. For Figure 4b, please provide more discussion on how 6TIC-4F reduces the non-radiative loss and charge transport loss.
9. As shown in SEM characterization, perovskite film with 6TIC-4F exhibits larger crystal size and dense morphology compared with control film. How does 6TIC-4F affect the kinetics of grain crystallization?

Reviewer 1

The manuscript by Wang et al. reports high efficiency all-inorganic perovskite solar cells enabled by defect passivation using a Lewis base 6TIC-4F. The passivation effect of 6TIC-4F is proved by the enhanced PL lifetime, improved Voc and EQE(EL) as well as reduced defect density calculated from the trap-filling limited voltage. The authors also tried to understand the passivation mechanism by theoretical simulation. The device performance of the solar cells based on the 6TIC-4F-treated perovskite film is indeed impressive compared to those of the reported inverted all-inorganic perovskite solar cells. Some questions before the publication of this ms:

1. In the title, “luminance” might not be the right word. Luminance is a measure of light output for visible LEDs with an unit of cd/m². However, there is no data showing the luminance of the solar cells in this manuscript.

Reply: In this manuscript, our fabricated solar cells possess good luminescence when they are operated in light-emitting diode mode. Therefore, we have revised the title as “**Highly Efficient and Luminescent All-inorganic Perovskite Solar Cells Enabled by Defect Passivation with 6TIC-4F Lewis Base**”

2. 6TIC-4F has many electron-rich functional groups such as -F, -C=O, -S and even the fused ring (Adv. Mater. 2018, 30, 1707583) can act as the passivation unit for perovskites. Experimental evidences (e.g., FTIR, XPS, or Raman) should be provided to further understand the passivation mechanism.

Reply: Considering the various spatial positions of electron-rich functional groups on 6TIC-4F, such as -F, -C=O, -CN, -S and even the fused ring (Adv. Mater., 2018, 30, 1707583), the steric effect of 6TIC-4F should be taken into consideration in analyzing potential passivation mechanisms. Before the theoretical simulation, we have employed the X-ray photoelectron spectroscopy (XPS) measurements to characterize the potential interactions between CsPbI_xBr_{3-x} and 6TIC-4F. As shown in **Figure S15**, the XPS feature peaks of Pb 4f and (N 1s, S 2p) showed obvious shift towards smaller and larger binding energy, respectively, after 6TIC-4F treatment, while the feature peaks of Cs 3d, F 1s and O 1s only shifted slightly. These XPS results indicated the potential interactions between Pb on CsPbI_xBr_{3-x} and -CN/-S groups on 6TIC-4F. Accordingly, we have conducted our DFT calculations on pristine Cs₅₆Pb₂₇I₁₀₈ cluster (**Figure S9**) and Pb exposed Cs₄₄Pb₂₇I₉₉ cluster (**Figure 3b**). To model the surface passivation effect of the 6TIC-4F molecule, we have sampled several docking positions of 6TIC-4F on the uncompensated Pb surface, allowing fully relaxation of the 6TIC-4F molecules. The 6TIC-4F surface density is about 0.003/Å². The optimized structures are shown in **Figure S8**. It can be observed that F-Pb, N-Pb, and S-Pb bonds stably formed after the geometry optimizations. The formation energies of the five motifs are -2.90 eV, -2.81 eV, -3.14 eV, -2.20 eV, and -3.61 eV, respectively, for **Figure S8a-e**. It is noteworthy that the S-Pb bonds can only be formed when the 6TIC-4F is on the edges of the grain boundaries due to the strong steric effect of the side chains. Therefore, after excluding the S-Pb bonding motif, the N-Pb bonding motif with a formation

energy of -3.14 eV is going to have a over 99% Boltzmann distribution ratio, which is likely to be the most effective electron-donating group to passivate most of the surface traps. This simulation is consistent with the XPS results that only the interaction between Pb and -CN/-S were detected.

The corresponding revision and figures in the manuscript and Supporting Information (yellow highlight) are listed below:

As shown in **Figure S15**, the XPS feature peaks of Pb 4f and (N 1s, S 2p) showed obvious shift towards smaller and larger binding energy, respectively, after 6TIC-4F treatment, while the feature peaks of Cs 3d, F 1s and O 1s only shifted slightly. These XPS results indicated the potential interactions between Pb on CsPbI_xBr_{3-x} and -CN/-S groups on 6TIC-4F. Accordingly, we have conducted our DFT calculations on pristine Cs₅₆Pb₂₇I₁₀₈ cluster (**Figure S9**) and Pb exposed Cs₄₄Pb₂₇I₉₉ cluster (**Figure 3b**). To model the surface passivation effect of the 6TIC-4F molecule, we have sampled several docking positions of 6TIC-4F on the uncompensated Pb surface, allowing full relaxation of the 6TIC-4F molecules, and then carried out the electronic structure computation. The 6TIC-4F surface density is about 0.003/Å². The optimized structures are shown in **Figure S8**. It can be observed that F-Pb, N-Pb, and S-Pb bonds stably formed after the geometry optimizations. The formation energies of the five motifs are -2.90 eV, -2.81 eV, -3.14 eV, -2.20 eV, and -3.61 eV, respectively, for **Figure S8a-e**. It is noteworthy that the S-Pb bonds can only be formed when the 6TIC-4F is on the edges of the grain boundaries due to the strong steric effect of the side chains. Therefore, after excluding the S-Pb bonding motif, the N-Pb bonding motif with a formation energy of -3.14 eV is going to have an over 99% Boltzmann distribution ratio, which is likely to be the most effective electron-donating group to passivate most of the surface traps. This simulation is consistent with the XPS results that only the interaction between Pb and -CN/-S were detected.

Figure S15. XPS feature spectra of Cs 1d (a), Pb 4f (b), F 1s (c), O 1s (d), N 1s (e) and S 2p (f) for 6TIC-4F film, CsPbI_xBr_{3-x} film and CsPbI_xBr_{3-x}/6TIC-4F film.

3. More details are suggested with regard to the photostability test. Are the devices kept working under light illumination during the photostability test? In addition, could the authors explain why the mixed halide perovskites used in this work are stable? There are many papers reporting phase segregation of mixed halide perovskites under light illumination (e.g., ACS Energy Lett. 2018, 31, 204-213; Chem. Sci., 2015, 6, 613-617; Nature Communications 2017, 8, 200).

Reply: We thank reviewer for the kind suggestion and nice comment. Our devices were not operated under continuous light illumination. In our photostability test, the interval measurements of device performance were conducted under continuous one sun equivalent illumination for 350 hours.

As reviewer mentioned, the phase segregation in organic-inorganic mixed-halide perovskite films was often observed in previous reports (*ACS Energy Lett.*, 2017, 2, 1416; *Nat. Commun.*, 2017, 8, 200; *Nat. Commun.*, 2018, 9, 4981). However, it has been reported that the phase segregation in inorganic CsPbI_xBr_{3-x} films was significantly suppressed under one sun illumination when the value of $x > 2.01$ (*J. Phys. Chem. Lett.* 2016, 7, 746). In our paper, the I/Br stoichiometry ratio in precursor solution is 4, meaning the x value of 2.4 in our CsPbI_xBr_{3-x}. Therefore, the phase segregation in our CsPbI_xBr_{3-x} films should also be suppressed. This phase stability has been demonstrated by the PL measurements of CsPbI_xBr_{3-x} films presented in **Figure S14**. The shape of PL spectra of CsPbI_xBr_{3-x} films with and without 6TIC-4F treatment only changed slightly under continuous one sun equivalent illumination for 30 min.

Furthermore, the surface defects have been shown to catalyze phase segregation in mixed-halide perovskites due to the carriers trapping and charge accumulation at perovskite surface (*ACS Energy Letters* 2018, 3, 2694). This indicates that the stability of our CsPbI_xBr_{3-x} films should be enhanced after 6TIC-4F passivation.

The corresponding revision and figures in the manuscript and Supporting Information (yellow highlight) are listed below:

It needs to be pointed out that the interval measurements of device performance were conducted under continuous one sun equivalent illumination for 350 h in our photostability test rather than keeping devices under continuous working condition.

In addition, it is worthy to note that the phase segregation in CsPbI_xBr_{3-x} films will be significantly alleviated when the value of x is > 2.01 .²³ Herein, the I/Br stoichiometry ratio in precursor solution (experimental section in Supporting Information) is 4, which means the x value is 2.4. Therefore, the phase segregation in our CsPbI_xBr_{3-x} films should also be suppressed like literature reported. In addition, the shape of PL spectra of CsPbI_xBr_{3-x} films with and without 6TIC-4F treatment only changed slightly under continuous one sun equivalent illumination for 30 min (**Figure S14**), demonstrating the good phase stability of our CsPbI_xBr_{3-x} films.

Furthermore, the surface defects have been shown to catalyze phase segregation in mixed-ion halide perovskites due to carriers trapping and charge accumulations at perovskite surface.⁴⁵ This indicates that the photostability of our CsPbI_xBr_{3-x} films should be enhanced after 6TIC-4F passivation.

Figure S14. The PL and normalized PL spectra of CsPbI_xBr_{3-x} films without (a, b) and with (c, d) 6TIC-4F treatment under continuous one sun equivalent illumination.

4. 6TIC-4F is introduced into the perovskite films by anti-solvent. Does 6TIC-4F function as well if it is introduced through perovskite precursor solution or post-treatment (e.g., spin-coating 6TIC-4F solution on top of annealed perovskite films)?

Reply: In **Figure S3**, the control devices post-treated with 6TIC-4F (Bilayer 6TIC-4F) were fabricated and compared with the control devices and devices based on 6TIC-4F treated films (Dripping of 6TIC-4F). The performance of the champion devices is listed in **Figure S16** and **Table S2**. The V_{OC} , J_{SC} , and FF of control device with bilayer 6TIC-4F increased to 1.14 V, 17.1 mA/cm² and 76.5%, respectively. Comparing the control device and the device treated with 6TIC-4F containing anti-solvent, it is clear that the passivation effect of 6TIC-4F still works for the control devices to improve the V_{OC} and FF, as well as the devices introduced 6TIC-4F by anti-solvent.

The corresponding revision and figures in the manuscript and Supporting Information (yellow highlight) are listed below:

In order to distinguish the effect of morphology on the improved performance, the control devices with 6TIC-4F post-treatment (Bilayer 6TIC-4F) were fabricated and compared with the control devices and devices based on 6TIC-4F-containing anti-solvent treated films. The performance of the champion devices is listed in **Figure S16** and **Table S2**. The V_{OC} , J_{SC} , and FF of the control device with bilayer 6TIC-4F increased to 1.14 V, 17.1 mA/cm² and 76.5%, respectively. Comparing the control device and device treated with 6TIC-4F containing anti-

solvent, it is clear that the major contribution from improved morphology is the increased J_{SC} , which may derive from the denser film. Therefore, the improved V_{OC} and FF may mainly come from the passivation effect of 6TIC-4F on $CsPbI_xBr_{3-x}$ films.

Figure S16. The JV curves of control device, control device with bilayer 6TIC-4F and device with dripping 6TIC-4F.

Table S2: The performance parameters of control device, control device with bilayer 6TIC-4F and device with dripping 6TIC-4F.

Devices	V_{oc} (V)	J_{sc} (mA/cm^2)	FF (%)	PCE (%)
Control	1.10	17.0	74.2	13.9
Dripping 6TIC-4F	1.16	17.7	78.6	16.1
Bilayer 6TIC-4F	1.14	17.1	76.5	14.9

5. In addition to the passivation effect, the enhanced film quality by 6TIC-4F could be another reason for the enhanced device performance. What is the mechanism of the enhanced grain size after 6TIC-4F treatment?

Reply: It is well known that the preparation of metal halide perovskite thin film is based on a sol-gel processing (*J. Mater. Chem. A*, 2016, 4, 8308). The solvates, such as $MAPbI_3 \cdot DMF$ and $MAPbI_3 \cdot DMSO$, formed during spin-coating process and could impact the morphology of perovskite film (*Adv. Mater.*, 2017, 29, 1604113; *Adv. Mater.*, 2019, 31, 1901284). Moreover, it has been reported that the nucleation and growth of the $MAPbI_3 \cdot DMSO$ solvate were promoted via introducing small molecules with Lewis base functional groups (*Adv. Mater.*, 2018, 30, 1706576). These large crystalline solvates would play a role as a template for further growth of perovskite when the solvent was removed, leading to the formation of perovskite phase. Therefore, we speculate that 6TIC-4F could facilitate the nucleation and growth of intermediate solvates when it was introduced into perovskite precursor during spin-coating process. These solvates would act as templates for further growth of perovskite, resulting in larger grain size and denser morphology than control film.

In **Figure S3**, the CsPbI_xBr_{3-x} film showed the larger grain size and the denser morphology after 6TIC-4F treatment, which could result in the fewer defects on grain boundaries (*Science*, 2015, 348, 1234; *Joule*, 2019, 3, 177) and less current leakage in devices (*Nat. Commun.*, 2019, 10, 4686; *Energy Environ. Sci.*, 2016, 9, 484). In order to distinguishing the effect of morphology on the performance improvement, the control devices with 6TIC-4F post-treatment (Bilayer 6TIC-4F) were fabricated and compared with the control devices and devices based on 6TIC-4F treated films (Dripping 6TIC-4F). The performance of champion devices is listed in **Figure S16** and **Table S2**. The V_{OC}, J_{SC} and FF of control device with bilayer 6TIC-4F increased to 1.14 V, 17.1 mA/cm² and 76.5%, respectively. Comparing to control device and device with dripping 6TIC-4F, it's clear that the major contribution of morphology improvement was the increase of J_{SC}, which may be beneficial to the denser film.

The corresponding revision in the manuscript (yellow highlight) is listed below:

It has been reported that the intermediate solvates (perovskite·solvent) formed during spin-coating process could impact the morphology of perovskite films.⁴⁰⁻⁴¹ The nucleation and growth of the solvates can be promoted by introducing small molecules with Lewis base functional groups.³¹ These large crystalline solvates could act as a template for perovskite growth when the solvent was removed, leading to the formation of perovskite phase. Therefore, we speculate that 6TIC-4F could facilitate the nucleation and growth of intermediate solvates when it was introduced into perovskite precursor solution during spin-coating process, and then these solvates would act as templates for further growth of perovskites, resulting in larger grain size and denser morphology than those of the control film.

In addition, the CsPbI_xBr_{3-x} film showed larger grain size and a denser morphology after the 6TIC-4F treatment (**Figure S3**), which could result in fewer defects on grain boundaries⁴⁶⁻⁴⁷ and less current leakage in devices.⁴⁸⁻⁴⁹ In order to distinguish the effect of morphology on the improved performance, the control devices with 6TIC-4F post-treatment (Bilayer 6TIC-4F) were fabricated and compared with the control devices and devices based on 6TIC-4F-containing anti-solvent treated films. The performance of the champion devices is listed in **Figure S16** and **Table S2**. The V_{OC}, J_{SC}, and FF of the control device with bilayer 6TIC-4F increased to 1.14 V, 17.1 mA/cm² and 76.5%, respectively. Comparing the control device and device treated with 6TIC-4F containing anti-solvent, it is clear that the major contribution from improved morphology is the increased J_{SC}, which may derive from the denser film. Therefore, the improved V_{OC} and FF may mainly come from the passivation effect of 6TIC-4F on CsPbI_xBr_{3-x} films.

Reviewer 2

In this work, Wang et. al present a defect passivation strategy for improving the performance of a Cs-based halide perovskite in the p-i-n device architecture. The study has been carefully conducted and the improvements in the resultant films and devices are impressive. There are, however, a few minor comments that I believe should be addressed before this paper is published.

1) The grammar and overall language in this article needs to be carefully checked for errors, as there are many of them which makes reading the paper a bit difficult at times. Also, redundancies such as "photo-illumination" need to be corrected.

Reply: We thank reviewer for the comments and kind suggestions. We have revised the paper thoroughly with corrections highlighted in red color in our manuscript and also listed below:

Original text	Revision
This work provides a new insight in the design of functional interlayers for improved efficiency and stability of all-inorganic PVSCs.	This work provides new insights in the design of functional interlayers for improving efficiencies and stability of all-inorganic PVSCs.
the instability issues under thermal and photo-illumination stresses are still needed to be addressed	the issues of instability under thermal and light illumination stresses still need to be addressed
Although the PCE of inorganic PVSCs still fall behind that of the organic-inorganic counterparts, these important works validate the promise of high efficiency CsPbI _x Br _{3-x} PVSCs.	Although the PCEs of inorganic PVSCs still fall behind those of the organic/inorganic counterparts, these important works validate the promise of CsPbI_xBr_{3-x} PVSCs.
It is noteworthy that most of high-performance inorganic PVSCs reported so far are based on the conventional n-i-p architecture	It is noteworthy that most of the high performance inorganic PVSCs reported so far are based on using the conventional n-i-p architecture
the performance of all these developed inverted inorganic PVSCs still significantly lag behind that of their conventional counterparts	the performance of all these inverted inorganic PVSCs developed are still significantly lagging behind those of their conventional counterparts
the abundant nitrogen (N) atoms with lone pair electrons on 6TIC-4F could passivat the surface defects of CsPbI _x Br _{3-x} film	the numerous nitrogen (N) atoms possessing lone pair electrons on 6TIC-4F could passivate the surface defects of CsPbI _x Br _{3-x} film
Afterwards, the films were first annealed at 55 °C for 50 s	Afterward , the films were first annealed at 55 °C for 50 s
films with 6TIC-4F treamtmet remained the diffraction patterns of perovskite with brighter spots	films with 6TIC-4F treatment have the diffraction patterns of perovskite remained with brighter spots
Afterwards, steady-state	Afterward , steady-state

photoluminescence (SSPL) and time-resolved photoluminescence (TRPL)	photoluminescence (SSPL) and time-resolved photoluminescence (TRPL)
possibly as a consequence of the reduced shallow defects on the CsPbI _x Br _{3-x} grain boundaries and surfaces	possibly as the consequence of reduced shallow defects on the CsPbI _x Br _{3-x} grain boundaries and surfaces
the maximum power point (MPP) in a N ₂ -filled glovebox	the maximum power point (MPP) in a nitrogen-filled glovebox
the photocurrent of the control device was gradually declined that might be due to the presence of a significant amount of trap states in the CsPbI _x Br _{3-x} film	the photocurrent of the control device gradually declined which might be due to the presence of significant trap states in the CsPbI _x Br _{3-x} film
This work provides a new insight about the mechanisms of trap passivation on inorganic perovskite surface	This work provides new insights of mechanism for surface trap passivation of inorganic perovskites

2) This paper is not very well referenced, particularly in the introduction. The introduction should be framed such that the work in this paper can be seen in proper context of the work that has already been done. The main focus of this work is passivation of the surfaces and grain boundaries with 6TIC-4F, and there is a large body of work that deals with the passivation of perovskite surfaces: lewis base passivation, passivation, as well as passivation using quarternary ammonium cations in both n-i-p and p-i-n solar cells. Work that has been done in this vein should at least be mentioned/referenced if not discussed. Additionally, for discussing the instability of dopants, the authors might consider citing the following papers (10.1002/aenm.201800554, 10.1039/C8TC02242A). The papers which the authors have referenced speak to the instability of Li-doped spiro but do not mention the problems with tbp.

Reply: We thank the reviewer very much for these helpful comments. We have mentioned more relevant works related to the surface passivation of perovskites in our manuscript. The two papers mentioned by the reviewer have also been cited in the manuscript.

The relevant revisions in the manuscript are highlighted in yellow and listed below:

However, severe degradation processes have been reported for devices using such doped HTLs due to the instability of dopants (e.g. 4-tert-butyl pyridine (TBP) and lithium salts).¹⁹⁻²²

It has been reported that the positively/negatively charged defects on perovskite surface and grain boundaries can induce large V_{OC} losses in PVSCs.²⁸⁻³⁰ To address this issue, numerous works have been performed to passivate the perovskite surface/grain boundaries, such as using Lewis base cations,³¹⁻³² polymer,^{16,33} and quaternary ammonium cations.³⁴ Inspired by these works, we present a simple molecular passivation strategy to reduce V_{OC} loss by reducing the density of surface defects in CsPbI_xBr_{3-x} film with π -conjugated 6TIC-4F, which can be

dissolved in anti-solvent (such as chlorobenzene, CB) to passivate uncoordinated defects on surface/grain boundaries.

Reviewer 3

In this manuscript, the authors reported a small organic molecule 6TIC-4F which assists crystallization of perovskite as well as charge transfer via suitable band alignment. The PCE of p-i-n structured PSCs has achieved 16.1% and a certificated value of 15.6%. This manuscript may be accepted by Nature Communications after the authors clearly address the following concerns.

1. Considering the multiple functional “-CN” groups, the large molecular size of 6TIC-4F and obvious enhancement of PV performance, it is important to provide evidence on how 6TIC-4F is distributed in the perovskite film and add relevant discussions in the maintext. How the charge carriers behavior at the grain boundaries and/or film surface based on 6TIC-4F distribution pattern? Does steric effect of 6TIC-4F also need to be taken into consideration in terms of “passivation”? One of the -CN groups in Figure S1a is missing. Please check.

Reply: We thank reviewer for the nice comments. According to previous reports, there is a perovskite – fullerene graded heterojunction exists on the film when the fullerene containing antisolvent was used during film preparation (*ACS Energy Lett.*, 2017, 2, 2531; *Nano Energy*, 2016, 30, 417; *Nat. Energy*, 2016, 1, 16148). Similar to the fullerene cases, we speculate that a perovskite – 6TIC-4F graded heterojunction should also be formed in our film, implying that 6TIC-4F had a gradient distribution in this heterojunction and the highest concentration should be on top of the film surface. These 6TIC-4F molecules should be located on surface and grain boundaries of perovskite film because they cannot be incorporated into the perovskite crystal frame due to its large molecular size.

As shown in **Figure 2d**, the LUMO of 6TIC-4F is lower than that of $\text{CsPbI}_x\text{Br}_{3-x}$ film, thus the photogenerated electrons should be transferred from perovskite to 6TIC-4F. Moreover, 6TIC-4F molecules can form highly crystalline packing (*Adv. Funct. Mater.*, 2018, 28, 1802324), which is beneficial for the intermolecular charge transfer. Therefore, the charge transport scenario can be described as the following: the electrons in the $\text{CsPbI}_x\text{Br}_{3-x}$ film could be extracted into 6TIC-4F layer then transported through the well packed 6TIC-4F molecules to afford improved J_{SC} and FF in PVSC.

Considering various spatial positions of the electron-rich functional groups on 6TIC-4F, such as -F, -C=O, -CN, -S and the fused ring (*Adv. Mater.*, 2018, 30, 1707583), the steric effect of 6TIC-4F should also be considered in analyzing possible passivation mechanisms. Before any theoretical simulations, we have conducted the X-ray photoelectron spectroscopy (XPS) measurements to analyze the potential interactions between $\text{CsPbI}_x\text{Br}_{3-x}$ and 6TIC-4F. As shown in **Figure S15**, the feature peaks of Pb 4f and (N 1s, S 2p) in XPS showed obvious shift towards smaller and larger binding energy, respectively, after the 6TIC-4F treatment, while the feature peaks of Cs 3d, F 1s and O 1s had only minor shift. These XPS results indicated some interactions between Pb on $\text{CsPbI}_x\text{Br}_{3-x}$ and -CN/-S groups on 6TIC-4F.

DFT calculations were also conducted on pristine $\text{Cs}_{56}\text{Pb}_{27}\text{I}_{108}$ cluster (**Figure S9**) and Pb exposed $\text{Cs}_{44}\text{Pb}_{27}\text{I}_{99}$ cluster (**Figure 3b**). We have sampled several docking positions of 6TIC-4F on the uncompensated Pb surface to model the surface passivation effect by the 6TIC-4F molecule. The 6TIC-4F surface density is about $0.003/\text{\AA}^2$. The optimized structures are shown in

Figure S8. It is shown that F-Pb, N-Pb, and S-Pb bonds can be stably formed after the geometry optimizations. The formation energies of the five motifs are -2.90 eV, -2.81 eV, -3.14 eV, -2.20 eV, and -3.61, respectively, for **Figure S8a-e**. It is noteworthy, the S-Pb bonds can only form when the 6TIC-4F is on the edges of the grain boundaries due to the strong steric effect of the side chains. Therefore, after excluding the S-Pb bonded motif, the N-Pb bonded motif with formation energy of -3.14 eV is going to have a distribution ratio over 99% according to the Boltzmann distribution, which is likely to be the most effective electron-donating group to passivate most of the surface traps. This simulation is consistent with the XPS results that only the interaction between Pb and -CN/-S were detected.

The error of missing -CN group in **Figure S1a** has been corrected.

The corresponding revision in the manuscript and Supporting Information (yellow highlight) is listed below:

Moreover, it has been reported that the perovskite – fullerene graded heterojunction could be formed within the perovskite film when the fullerene containing anti-solvent was used during the film preparation.³⁷⁻³⁹ We speculate that a perovskite – 6TIC-4F graded heterojunction should also be formed within the $\text{CsPbI}_x\text{Br}_{3-x}$ film similar to those of the fullerene cases. This implies that there is a gradient distribution of 6TIC-4F within the heterojunction and the highest concentration should be on top of the film surface. In addition, due to the large size of the 6TIC-4F molecules, they should be located on the surface and grain boundaries of $\text{CsPbI}_x\text{Br}_{3-x}$ film, because they cannot be incorporated into the perovskite crystalline frame.

Electrons can then be extracted from the $\text{CsPbI}_x\text{Br}_{3-x}$ film into the 6TIC-4F molecules. Due to the tendency for 6TIC-4F to form good packing,³⁶ it will facilitate electron transport and collection for 6TIC-4F treated $\text{CsPbI}_x\text{Br}_{3-x}$ PVSC to result in higher J_{SC} and FF.

As shown in **Figure S15**, the XPS feature peaks of Pb 4f and (N 1s, S 2p) showed obvious shift towards smaller and larger binding energy, respectively, after 6TIC-4F treatment, while the feature peaks of Cs 3d, F 1s and O 1s only shifted slightly. These XPS results indicated the potential interaction between Pb on $\text{CsPbI}_x\text{Br}_{3-x}$ and -CN/-S groups on 6TIC-4F. Then, DFT calculations were conducted on pristine $\text{Cs}_{56}\text{Pb}_{27}\text{I}_{108}$ cluster (**Figure S9**) and Pb exposed $\text{Cs}_{44}\text{Pb}_{27}\text{I}_{99}$ cluster (**Figure 3b**). To model the surface passivation effect of the 6TIC-4F molecule, we have sampled several docking positions of 6TIC-4F on the uncompensated Pb surface, allowing full relaxation of the 6TIC-4F molecules, and then carried out the electronic structure computation. The 6TIC-4F surface density is about $0.003/\text{\AA}^2$. The optimized structures are shown in **Figure S8**. It can be observed that F-Pb, N-Pb, and S-Pb bonds stably formed after the geometry optimizations. The formation energies of the five motifs are -2.90 eV, -2.81 eV, -3.14 eV, -2.20 eV, and -3.61 eV, respectively, for **Figure S8a-e**. It is noteworthy that the S-Pb bonds can only be formed when the 6TIC-4F is on the edges of the grain boundaries due to the strong steric effect of the side chains. Therefore, after excluding the S-Pb bonding motif, the N-Pb bonding motif with a formation energy of -3.14 eV is going to have an over 99% Boltzmann distribution ratio, which is likely to be the most effective electron-donating group to passivate most of the surface traps. This simulation is consistent with the XPS results that only the interaction between Pb and -CN/-S were detected.

A $\text{Cs}_{56}\text{Pb}_{27}\text{I}_{108}$ cluster has been extracted from the optimized α - CsPbI_3 supercell.

Figure 3. Theoretical simulation. (a) Illustration of the perovskite and 6TIC-4F structures for DFT calculation. (b) The valence electron density of the Pb exposed to $\text{Cs}_{44}\text{Pb}_{27}\text{I}_{99}$ cluster. (c) J - V characteristics of devices with ITO/SnO₂/CsPbI_xBr_{3-x}/ZnO/Ag configuration for estimating the defect density in the films. (d) Illustration of possible passivation mechanism and potential interaction sites. (e) The valence electron density of the most favorable 6TIC-4F/ $\text{Cs}_{44}\text{Pb}_{27}\text{I}_{99}$ motif. (f) The energy-level diagram of the trap-states passivation by 6TIC-4F.

Figure S8. The optimized structures of docking positions of 6TIC-4F on the uncompensated Pb surface, allowing fully relaxation of the 6TIC-4F molecules. The 6TIC-4F surface density is about $0.003/\text{\AA}^2$. Only F-Pb, N-Pb and S-Pb bonds stably formed after the geometry optimization. The formation energies of the five motifs are -2.90 eV, -2.81 eV, -3.14 eV, -2.20 eV, and -3.61 eV respectively for **Figure S8a-e**.

Figure S9. The valence electron density of the Pb compensated pristine $\text{Cs}_{56}\text{Pb}_{27}\text{I}_{108}$ cluster.

Figure S15. XPS feature spectra of Cs 1d (a), Pb 4f (b), F 1s (c), O 1s (d), N 1s (e) and S 2p (f) for 6TIC-4F film, $\text{CsPbI}_x\text{Br}_{3-x}$ film and $\text{CsPbI}_x\text{Br}_{3-x}/6\text{TIC-4F}$ film.

Figure S1. Illustration of chemical structure of 6TIC-4F (a) and the V_{OC} and PCE of recently published all-inorganic PVSCs based on inverted structure in this work and in the literatures (b) (details in Table S1).

2. Page 5 line 133, “by dropping 130 mL solution of 6TIC-4F in chlorobenzene”, should be μL . In addition, the authors describe the “Device Fabrication” procedure in the supporting information as “the perovskite active layer was prepared by a widely used two-step spin-coating procedure at 1500 rpm (for 10 s) and 5000 rpm (for 30 s). When the spin-coating reached 5000 rpm, 0.2 ml 6TIC-4F (3 mg/ml, CB) was dripped onto the substrates.” Please be consistent with those expressions in the maintext, e. g. one-step or two-step? 0.2 ml or 130 μL ? ml or mL? Fabrication of the control film should also be provided.

Reply: We thank reviewer for the kind comments. We have revised the device fabrication procedure and added the fabrication of control film in Supporting Information. Meanwhile, we have revised “130 mL solution of 6TIC-4F” to be “130 μL solution of 6TIC-4F” and made the description of device fabrication consistent in the manuscript and in the Supporting Information.

The corresponding revision in the manuscript and Supporting Information (yellow highlight) is listed below:

by one-step spin-coating method by adding dropwise 130 μL solution of 6TIC-4F in chlorobenzene (CB) (3 mg/mL) onto the precursor film

the perovskite active layer was prepared by the commonly used one-step spin-coating procedure at 1500 rpm (for 10 s) and 5000 rpm (for 30 s). When the spin-coating speed reached 5000 rpm, 0.2 mL CB or 6TIC-4F in CB (3 mg/mL) was dripped onto the substrates for control film or film with 6TIC-4F, respectively.

3. Page 6, line 135, the authors state that “followed by annealing at 255 °C for another 50 s to get the black phase perovskite”, please provide TGA measurement for 6TIC-4F both of its pure form and with perovskite.

Reply: We have conducted the TGA measurements for 6TIC-4F and perovskite with 6TIC-4F treatment. The results and the corresponding discussion have been shown in **Figure S11** in Supporting Information and in the manuscript with yellow highlight, respectively.

The corresponding revision and figures in the manuscript and Supporting Information (yellow highlight) were listed below:

It needs to be pointed out that the thermal gravimetric analysis (TGA) results in **Figure S11** showed that neither 6TIC-4F nor 6TIC-4F/ $\text{CsPbI}_x\text{Br}_{3-x}$ decompose in the range between 200 to 300 °C, indicating that the annealing condition in this work (255 °C) did not damage 6TIC-4F.

Figure S11. The thermal gravimetric analysis (TGA) curves of 6TIC-4F and perovskite/6TIC-4F from room temperature to 600 °C.

4. Please provide film thickness when comparing the absorbance in Figure 1d.

Reply: We thank the reviewer for the nice suggestion. Both the thickness of CsPbI_xBr_{3-x} films with and without 6TIC-4F treatment are ~ 350 nm measured by DektakXT Profiler (Bruker).

The corresponding revision in the manuscript and Supporting Information (yellow highlight) are listed below:

Presented in **Figure 1d** is the UV-vis absorbance of both films with a thickness of ~ 350 nm. The thicknesses of films were measured by using a DektakXT Profiler (Bruker).

5. Characterizations such as XPS should be provided to show the existence of Pb²⁺ or metallic Pb defects. Apparently the authors only discussed the under coordinated Pb²⁺ type of defect, does the film morphology also impacts the PV performance, and why? Please add discussion.

Reply: We thank the reviewer for these nice comments. The states of Pb in CsPbI_xBr_{3-x} film have been studied via the X-ray photoelectron spectroscopy (XPS) and shown in **Figure S12**. According to the XPS results, the Pb 4f_{7/2} core level can be deconvoluted into two peaks located at 136.8 eV and 138.1 eV, respectively. The intensity of Pb²⁺ (138.1 eV) was significantly higher than that of Pb⁰ (136.8 eV) which indicated the existence of Pb²⁺ defects.

In **Figure S3**, the CsPbI_xBr_{3-x} film showed the larger grain size and a denser morphology after the 6TIC-4F treatment, which could result in fewer defects on grain boundaries (*Science*, 2015, 348, 1234; *Joule*, 2019, 3, 177) and less current leakage in devices (*Nat. Commun.*, 2019, 10, 4686; *Energy Environ. Sci.*, 2016, 9, 484). In order to distinguish the effect of morphology on the improved performance, the control devices with 6TIC-4F post-treatment (Bilayer 6TIC-4F) were fabricated and compared with the control devices and devices based on 6TIC-4F treated films (Dripping 6TIC-4F). The performance of the champion devices are listed in **Figure S16**

and **Table S2**. The V_{OC} , J_{SC} , and FF of control device with bilayer 6TIC-4F increased to 1.14 V, 17.1 mA/cm² and 76.5%, respectively. Comparing the control device and the device with 6TIC-4F treatment, it is clear that the major contribution of improved morphology is the J_{SC} increase, which may be due to the denser film.

The corresponding revision and figures in the manuscript and Supporting Information (yellow highlight) were listed below:

In order to verify the potential interactions between CsPbI_xBr_{3-x} and 6TIC-4F, X-ray photoelectron spectroscopy (XPS) measurements were conducted to characterize the elemental states on the CsPbI_xBr_{3-x} surface before and after the 6TIC-4F treatment. **Figure S12** presented the states of Pb of the CsPbI_xBr_{3-x} film and showed two feature peaks of Pb 4f_{7/2} and Pb 4f_{5/2}. The Pb 4f_{7/2} core level can be deconvoluted into two peaks located at 136.8 eV and 138.1 eV, respectively. The intensity of Pb²⁺ (138.1 eV) was significantly higher than that of Pb⁰ (136.8 eV) and indicated the existence of Pb²⁺ defects.

In addition, the CsPbI_xBr_{3-x} film showed larger grain size and a denser morphology after the 6TIC-4F treatment (**Figure S3**), which could result in fewer defects on grain boundaries⁴⁶⁻⁴⁷ and less current leakage in devices.⁴⁸⁻⁴⁹ In order to distinguish the effect of morphology on the improved performance, the control devices with 6TIC-4F post-treatment (Bilayer 6TIC-4F) were fabricated and compared with the control devices and devices based on 6TIC-4F-containing anti-solvent treated films. The performance of the champion devices is listed in **Figure S16** and **Table S2**. The V_{OC} , J_{SC} , and FF of the control device with bilayer 6TIC-4F increased to 1.14 V, 17.1 mA/cm² and 76.5%, respectively. Comparing the control device and device treated with 6TIC-4F containing anti-solvent, it is clear that the major contribution from improved morphology is the increased J_{SC} , which may derive from the denser film. Therefore, the improved V_{OC} and FF may mainly come from the passivation effect of 6TIC-4F on CsPbI_xBr_{3-x} films.

Figure S12. The XPS signals of Pb states in CsPbI_xBr_{3-x} film.

Figure S16. The JV curves of control device, control device with bilayer 6TIC-4F and device with dripping 6TIC-4F.

Table S2: The performance parameters of control device, control device with bilayer 6TIC-4F and device with dripping 6TIC-4F.

Devices	V_{oc} (V)	J_{sc} (mA/cm^2)	FF (%)	PCE (%)
Control	1.10	17.0	74.2	13.9
Dripping 6TIC-4F	1.16	17.7	78.6	16.1
Bilayer 6TIC-4F	1.14	17.1	76.5	14.9

6. It would be fair to prepare control film with high quality (pinhole free) to make comparison rather than that shown in Figure S3a as there are already many mature methods with which such films can be achieved.

Reply: We thank reviewer for this kind comment. In the previous reports about inorganic perovskite solar cells (PVSCs), most of the $\text{CsPbI}_x\text{Br}_{3-x}$ films were prepared without anti-solvent washing (*Nat. Commun.*, 2018, 9, 2225; *Nat. Commun.*, 2018, 9, 4544; *Nat. Commun.*, 2019, 10, 4686; *Joule*, 2019, 3, 205; *ACS Energy Lett.*, 2018, 3, 970; *Adv. Mater.*, 2019, 31, 1901152; *Adv. Mater.* 2019, 1905143). However, in this work, we employed 6TIC-4F containing chlorobenzene (CB) anti-solvent to passivate $\text{CsPbI}_x\text{Br}_{3-x}$ films, thus our control film should be prepared with the same procedure in CB without 6TIC-4F. The morphology of our control film (shown in **Figure S3a**) is the optimal result in our current experiments, and it is quite comparable to those control films prepared without anti-solvent (*Adv. Mater.* 2019, 1905143; *Joule*, 2019, 3, 205; *ACS Energy Lett.*, 2018, 3, 970; *Adv. Mater.* 2019, 31, 1900605) and with anti-solvent (*Angew.Chem.Int.Ed.*2018,57,12745–12749; *Joule*, 2019, 3, 191) in previous reports. As the reviewer mentioned, the control film with higher quality could be achieved with further optimization, however, it is beyond the focus of this work and will be studied in the near future.

7. Please give reasons for enhancing the stability of PSCs after 6TIC-4F treatment.

Reply: It has been well known that the charged defects on the surface of perovskite film and/or grain boundaries have lower reaction activation energy (*Nat. Energy*, 2018, 3, 648), which can result in the device decay more easily due to the reaction with moisture, oxygen or light. Therefore, the 6TIC-4F passivation should prevent the defects from these attacks. In addition, the contact angle measurements were conducted to study the wetting behavior of CsPbI_xBr_{3-x} film surface with and without 6TIC-4F treatment. In **Figure S13**, the contact angle of CsPbI_xBr_{3-x} film surface increased from 34.6° to 40.1° after 6TIC-4F treatment, implying a slightly more hydrophobic surface. These results showed that the 6TIC-4F passivation should be able to improve the stability of CsPbI_xBr_{3-x} based PSCs because of the enhanced resistance for moisture, oxygen or light. This clearly shows the advantages of the 6TIC-4F-treated film.

Furthermore, the surface defects have been shown to facilitate phase segregation in mixed-ion halide perovskite due to the carriers trapping and charge accumulation at perovskite surface (*ACS Energy Letters* 2018, 3, 2694). This indicates that the phase stability of our CsPbI_xBr_{3-x} films could also be enhanced after 6TIC-4F passivation.

The corresponding revision in the manuscript (yellow highlight) is listed below:

It is known that the charged defects on perovskite surface and/or grain boundaries have lower reaction active energy, which more easily results in the degradation of perovskite film under the attacks of moisture, oxygen or light.⁴⁴ Therefore, the 6TIC-4F passivation should prevent the defects from these attacks. In addition, as shown in **Figure S13**, the contact angle of CsPbI_xBr_{3-x} film surface increased from 34.6° to 40.1° after 6TIC-4F treatment, implying a more hydrophobic film with 6TIC-4F. These results showed that 6TIC-4F passivation should improve the stability of CsPbI_xBr_{3-x} based PSCs because of the enhanced resistance for moisture, oxygen or light. This clearly shows the advantages of the 6TIC-4F-treated film.

Furthermore, the surface defects have been shown to catalyze phase segregation in mixed-ion halide perovskites due to carriers trapping and charge accumulations at perovskite surface.⁴⁵ This indicates that the photostability of our CsPbI_xBr_{3-x} films should be enhanced after 6TIC-4F passivation

Figure S13. Isopropanol contact-angle measurements of CsPbI_xBr_{3-x} films without (a) and with (b) 6TIC-4F treatment.

8. For Figure 4b, please provide more discussion on how 6TIC-4F reduces the non-radiative loss and charge transport loss.

Reply: In **Figure 3**, our DFT theoretical simulation results showed that the -CN groups on the 6TIC-4F might be the most possible electron-donating part to passivate the surface traps of perovskite film. Thus, the reduced non-radiative loss should be due to this passivation effect.

As mentioned previously in Comment 1, 6TIC-4F has the appropriate energy alignment between $\text{CsPbI}_x\text{Br}_{3-x}$ film and ZnO layer. In addition, it also has good electron-transporting property due to the highly crystalline packing. These features benefit the charge collection and reduce the charge transport loss.

The corresponding revision in the manuscript (yellow highlight) is listed below:

As illustrated in **Figure 3**, the -CN groups on the 6TIC-4F can passivate the surface traps of perovskite film and reduce the non-radiative loss. Meanwhile, the improved charge transfer in $\text{CsPbI}_x\text{Br}_{3-x}$ film with 6TIC-4F treatment benefits the charge collection, which has been discussed above, and thus reduce the charge transport loss. As shown in **Figure 4b**, the non-radiative loss in the 6TIC-4F-treated device was clearly suppressed, while the charge transport loss was also slightly decreased. These results verified that 6TIC-4F passivation not only suppresses the non-radiative recombination in the device, but also improves charge transport in the device.

9. As shown in SEM characterization, perovskite film with 6TIC-4F exhibits larger crystal size and dense morphology compared with control film. How does 6TIC-4F affect the kinetics of grain crystallization?

Reply: It is well known that the preparation of metal halide perovskite thin film is similar to the sol-gel processing (*J. Mater. Chem. A*, 2016, 4, 8308). The solvates, such as $\text{MAPbI}_3 \cdot \text{DMF}$ and $\text{MAPbI}_3 \cdot \text{DMSO}$ formed during the spin-coating process, could impact the morphology of perovskite film (*Adv. Mater.*, 2017, 29, 1604113; *Adv. Mater.*, 2019, 31, 1901284). Moreover, it has been reported that the nucleation and growth of the $\text{MAPbI}_3 \cdot \text{DMSO}$ solvate were promoted via introducing small molecules with Lewis base functional groups (*Adv. Mater.*, 2018, 30, 1706576). These large crystalline solvates could play the role as a template for further growth of perovskite when the solvent was removed, leading to the formation of perovskite phase. Therefore, we speculate that 6TIC-4F could facilitate the nucleation and growth of intermediate solvates when it was introduced into perovskite precursor during the spin-coating process. These solvates would act as templates for further growth of perovskites, resulting in larger grain size and denser morphology than control film.

The corresponding revision in the manuscript (yellow highlight) is listed below:

It has been reported that the intermediate solvates (perovskite·solvent) formed during spin-coating process and could impact the morphology of perovskite film.⁴⁰⁻⁴¹ The nucleation and growth of the solvates can be promoted by introducing small molecules with Lewis base

functional groups.³¹ These large crystalline solvates could act as a template for perovskite growth when the solvent was removed, leading to the formation of perovskite phase. Therefore, we speculate that 6TIC-4F could facilitate the nucleation and growth of intermediate solvates when it was introduced into perovskite precursor solution during spin-coating process, and then these solvates would act as templates for further growth of perovskites, resulting in larger grain size and denser morphology than those of the control film.

Meanwhile, the employed 6TIC-4F tends to trigger nucleation of perovskite precursor, leading to the formation of larger grain size and denser film.

REVIEWERS' COMMENTS:

Reviewer #1 (Remarks to the Author):

The authors have carefully addressed my concerns. I hence suggest to accept the ms as is.

Reviewer #2 (Remarks to the Author):

The authors have, in my view, adequately addressed the concerns of all reviewers. I recommend this paper for publication in its current form.

Reviewer #3 (Remarks to the Author):

The authors have addressed all my concerns and I recommend its publication in Nature Communications

Reviewer's Comments:

Reviewer #1 (Remarks to the Author):

The authors have carefully addressed my concerns. I hence suggest to accept the ms as is.

Reply: We are pleasure that our replies addressed the reviewer's concerns and thank reviewer for the affirmation. We also appreciate the kind and constructive comments from reviewer for improving our work.

Reviewer #2 (Remarks to the Author):

The authors have, in my view, adequately addressed the concerns of all reviewers. I recommend this paper for publication in its current form.

Reply: We are pleasure that our replies addressed the reviewer's concerns and thank reviewer for the affirmation. We also appreciate the kind and constructive comments from reviewer for improving our work.

Reviewer #3 (Remarks to the Author):

The authors have addressed all my concerns and I recommend its publication in Nature Communications

Reply: We are pleasure that our replies addressed the reviewer's concerns and thank reviewer for the affirmation. We also appreciate the kind and constructive comments from reviewer for improving our work.